# Learning Symmetric Representations for Equivariant World Models

## Abstract

Incorporating symmetries leads to highly data-efficient and generalizable world models. However, characterizing how underlying symmetries manifest in the input space is often difficult. We provide a method to map an input space (e.g. images) where we do not know the effect of transformations onto a feature space that transforms in a known manner under these operations. Specifically, we consider equivariant transition models as an inductive bias for learning the encoder. Our method allows existing equivariant neural networks to operate in previously inaccessible domains. We evaluate the effectiveness of this approach in domains with 3 distinct forms of underlying symmetry. In many cases, we demonstrate improvements relative to both fully-equivariant and non-equivariant baselines.

## 1 Introduction

Symmetry has proved to be a powerful inductive bias for improving generalization in supervised and unsupervised learning. A symmetry group defines an equivalence class of inputs in terms of a set of transformations that can be performed on this input, along with corresponding transformations for the output. The last years have seen many proposed equivariant models that incorporate symmetries into deep neural networks (Cohen & Welling, 2016a;b; Cohen et al., 2019; Weiler & Cesa, 2019; Weiler et al., 2018; Kondor & Trivedi, 2018; Bao & Song, 2019; Worrall et al., 2017). This results in models that are often more parameter efficient, more sample efficient, and safer to use by behaving more consistently in new environments.

However, a major impediment to the wider application of equivariant models is that it is not always obvious how a symmetry group acts on an input data set. As an example, let us consider the two pairs of images in Figure 1. On the left, we have a pair of MNIST digits, for which a 2D rotations in pixel space should induce a corresponding rotation in feature space. Here it is possible to achieve state-of-the-art accuracy using an $E(2)$-equivariant network (Weiler & Cesa, 2019). By contrast, exploiting the underlying symmetry is much more challenging for the pair of images on the right, which show the same three-dimensional object in two orientations. For these images, there is also an underlying symmetry group of rotations, but it is not easy to characterize the transformation in pixel space that is associated with a particular rotation.

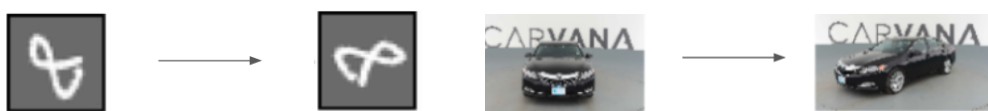

Figure 1: On MNIST, the rotation is easy to compute, allowing for the use of equivariant models. The rotation is difficult to compute for the car, making it harder to apply equivariant methods (Carvana, 2017).

In this paper, we consider the task of learning symmetric representations of data in domains where transformations cannot be hard-coded, which is to say that the *group action* on the data is not known. We propose training a standard network to learn a mapping from an input space, for which the group action is difficult to characterize, into a latent space, for which the action is known. This mapping, which we refer to as a symmetric embedding network, can then be composed with any equivariant network for downstream predictions.

As a concrete instantiation of this idea, we focus on learning world models, i.e. models that encode the effects of actions in the state space of an environment. We combine the symmetric embedding network with an equivariant transition network, which are trained end-to-end by minimizing a contrastive loss. The underlying intuition behind this approach is that the symmetry group of the transition model acts as an inductive bias that guides the embedding network to a representation that is as equivariant as possible. At the same time, incorporating symmetries into the transition dynamics has the potential to improve both data efficiency and out-of-distribution generalization.

The idea of learning symmetric embeddings using a mix of standard and equivariant networks has, to our knowledge, not previously been proposed or demonstrated. However, note that we are not proposing a new equivariant neural network design. In fact, our approach is useful precisely because it can be paired with any existing equivariant neural network to extend its applicability to new domains with unknown group actions. We apply our method to 5 different domains, 3 different symmetry groups, and 3 different equivariant architectures.

We summarize our contributions as follows:

- We introduce a meta-architecture for equivariant world models with symmetric embedding networks, which can be trained end-to-end using triples from a replay buffer by minimizing a contrastive loss.

- We demonstrate that this meta-architecture can be used to learn world models with a variety of equivariances in a self-supervised manner.

- Moreover, we show that models that have been trained using only a subset of all input actions can generalize to unseen input actions at test time.

## 2 RELATED WORK

**Equivariant Neural Networks**    A multitude of equivariant neural networks have been devised to impose various symmetry groups with respect to various groups across a variety of data types. They require that the group $G$ is known and the the group action on input, output, and hidden spaces is explicitly constructed. Examples include $G$-convolution (Cohen & Welling, 2016a), $G$-steerable convolution (Cohen & Welling, 2016b; Weiler & Cesa, 2019), tensor product and Clebsh-Gordon decomposition (Thomas et al., 2018), or convolution in the Fourier domain (Esteves et al., 2017). They have been applied to many data types such as gridded data (Weiler & Cesa, 2019), spherical data (Cohen et al., 2018), point clouds (Dym & Maron, 2020), and sets (Maron et al., 2020). They have found applications in many domains including molecular dynamics (Anderson et al., 2019), particle physics (Bogatskiy et al., 2020), and trajectory prediction (Walters et al., 2020). In particular, Ravindran (2004) consider symmetry in the context of Markov Decision Processes (MDPs) and van der Pol et al. (2020b) construct equivariant policy networks for policy learning. Our work also considers MDP with symmetry but focuses on learning equivariant world models (see Appendix B).

**Learning Symmetry**    Our work occupies a middle ground between equivariant neural networks in which the group and its representations are known and symmetry discovery models. Symmetry discovery methods attempt to learn both the group and actions from data. For example, Zhou et al. (2020) learn equivariance by learning a parameter sharing scheme using meta-learning. Dehmamy et al. (2021) similarly learn a basis for a Lie algebra generating a symmetry group while simultaneously learning parameters for the equivariant convolution over this symmetry group. Benton et al. (2020) propose an adaptive data augmentation scheme, where they learn which group of spatial transformations best supports data augmentation.

Higgins et al. (2018) define disentangled representations based on symmetry, with latent factors considered disentangled if they are independently transformed by commuting subgroups. Within this definition, Quessard et al. (2020) learn the underlying symmetry group by interacting with the environment, where the action space is a group of symmetry transformations. Except for the teapot task, we handle the more general case where the action space may be different from the symmetry group. Their latent transition is given by multiplication with a group element, whereas our transition model is given by an equivariant neural network.

**Structured Latent World Models**   World models learn state representations by ignoring unnecessary information unrelated to predicting environment dynamics. Such models are frequently used for high-dimensional image inputs, and usually employ (1) reconstruction loss (Ha & Schmidhuber, 2018; Watter et al., 2015; Hafner et al., 2019; 2020) or (2) constrastive loss. Contrastive loss is known to be less computationally costly and can produce good representations in learning from high-dimensional inputs (Oord et al., 2018; Anand et al., 2019; Chen et al., 2020; Srinivas et al., 2020; van der Pol et al., 2020a), thus we use it for training our models. For example, Kipf et al. (2020) learn object factored representations for structured world modeling with GNNs, which respect $S_n$ permutation symmetry. We learn symmetric representations for groups $G$ and explicitly enforce $G$-equivariance constraints to latent transition networks.

## 3   BACKGROUND

We provide some background on symmetry groups and highlight the difference between abstract symmetry groups and their concrete representations. Here we assume we know the abstract group, but only some of its relevant representations.

**Groups, Actions, and Representations**   A symmetry *group* consists of a set $G$ together with a composition map $\circ\colon G \times G \to G$. The group must contain an identity $1 \in G$ and each element $g \in G$ must be invertible. An action of the group $G$ on a set $S$ is a map $a\colon G \to \mathrm{Perm}(S)$ mapping each element of the group $g$ to a permutation $\pi_g \in \mathrm{Perm}(S)$ of the elements of $S$. Composition of group elements is compatible with the action such that $a(g_1 g_2, s) = a(g_1, a(g_2, s))$ for $g_1, g_2 \in G, s \in S$. A real *representation* of the group $G$ is a linear group action, given by a map $\rho\colon G \to \mathrm{GL}_n(\mathbb{R})$ which maps each element of $G$ to an invertible $n \times n$ matrix. The multiplication table of these matrices must match that of the abstract group elements under composition. That is, $\rho(g_1 \circ g_2) = \rho(g_1)\rho(g_2)$. See Hall (2003) for additional background on groups and their representations.

**Equivariant and Invariant Functions**   Given a function $f\colon X \to Y$ between vector spaces $X$ and $Y$ and a group $G$ equipped with representations $\rho_X$ and $\rho_Y$ acting on $X$ and $Y$ respectively, we say that $f$ is *equivariant* if, for all $x \in X, g \in G$, we have $f(\rho_X(g) \cdot x) = \rho_Y(g) \cdot f(x)$. This means that if the input is transformed by $g$ the output will be transformed correspondingly. The composition of equivariant functions is equivariant. Thus we can model equivariant functions using equivariant neural networks which alternate equivariant linear layers and equivariant non-linearities.

## 4   SYMMETRIC EMBEDDINGS FOR EQUIVARIANT WORLD MODELS

Our goal is to learn equivariant world models where the underlying symmetry group of the input space is difficult to compute. Our method gives a general template that can be fit to different equivariant neural networks, symmetries, and data types. We describe the general template and the implementation in examples.

### 4.1   MODEL OVERVIEW

**Equivariant World Models**   Let $\mathcal{S}$ be the state space and $\mathcal{A}$ be the action space of an environment. We consider a deterministic transition function $T : \mathcal{S} \times \mathcal{A} \to \mathcal{S}$ which outputs the next state $s' = T(s, a)$ given state $s$ and action $a$. Our goal is to learn $T$ from tuples $(s, a, s')$ collected from offline data. As the space $\mathcal{S} \times \mathcal{A}$ is combinatorially large, we wish to learn a compact state representation of the state and an accurate model of $T$ that can generalize to unseen transitions. This is accomplished by learning a state abstraction map $\mathcal{S} \to \mathcal{Z}$ and then learning transitions in latent space $T_{\mathcal{Z}}\colon \mathcal{Z} \times \mathcal{A} \to \mathcal{Z}$.

To learn world models from fewer samples, we exploit inherent symmetries of the environment. Let $G$ be a group of symmetries with group representations $\rho_{\mathcal{S}}$ and $\rho_{\mathcal{A}}$. We assume that $\rho_{\mathcal{A}}(g)$ is independent of state for simplicity as many domains do not require state-dependent actions. Additional details on the symmetric MDPs and the setup are provided in the Appendix. The transition function $T$ is equivariant if $T(\rho_{\mathcal{S}}(g) \cdot s, \rho_{\mathcal{A}}(g) \cdot a) = \rho_{\mathcal{S}}(g) \cdot T(s, a)$. We wish to enforce this on a neural network model for $T$. If trained to predict $s' = T(s, a)$, the model will then automatically generalize to $gs' = T(gs, ga)$, enabling improved generalization and sample efficiency.

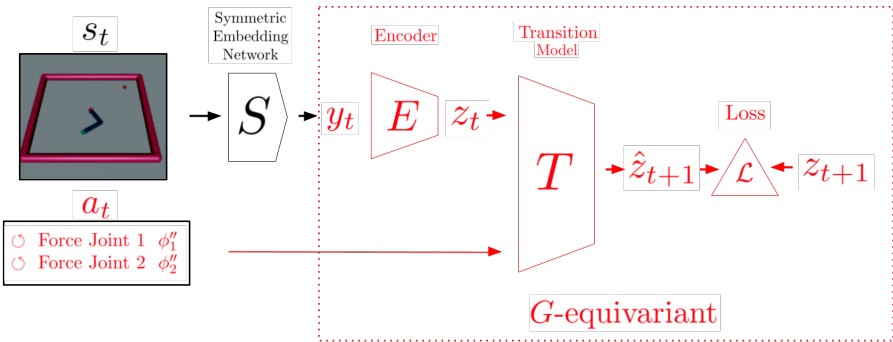

Figure 2: Diagram of the model architecture of our $G$-equivariant world model. The features in red have an explicit $G$-action $\rho$. The networks in red are $G$-equivariant. The example *Reacher* input has $G = D_4$ symmetry. The MDP actions have $G$-representation type $\rho_{\text{flip}}$ meaning they are reversed in sign by reflections and unaltered by rotations. The Symmetric Embedding Network is a CNN and the Encoder and Transition model are $E(2)$-CNNs with fiber group $D_4$.

Current methods for constructing model classes of equivariant neural networks require that $\rho_S$ and $\rho_A$ are known. In our case, however, although we assume the symmetry group $G$ and group action $\rho_A$, *we assume that $\rho_S$ is not explicitly known.* In many environments, $s$ is a pixel-level input and the transformation $\rho_S$ may be very difficult to describe. Thus, we cannot directly enforce symmetry for $T$ using an equivariant neural network.

**Meta-Architecture**    To learn an equivariant world model without access to $\rho_S$, we learn a symmetric abstract state mapping from states $s$ to abstract states $z$ in a space $\mathcal{Z}$ with an explicit action $\rho_{\mathcal{Z}}$ of the symmetry group $G$. We then learn a transition model in latent space where we can enforce symmetry using an equivariant neural network.

We learn the symmetric abstract state mapping in two parts. First we map the pixel-space state to an intermediate space $\mathcal{Y}$ using a *symmetric embedding $S\colon \mathcal{S} \to \mathcal{Y}$. This is a non-equivariant neural network acts as a feature extractor and maps images into a space $\mathcal{Y}$ with a simpler explicit symmetry group action $\rho_{\mathcal{Y}}$. We then map the intermediate space to the lower-dimensional latent space using an *equivariant encoder $E\colon \mathcal{Y} \to \mathcal{Z}$. The encoder discards features that are unnecessary to predict dynamics and reduces dimensionality. Lastly, we compute the *transition* in latent space, $T\colon \mathcal{Z} \times \mathcal{A} \to \mathcal{Z}$. We explicitly enforce $E$ and $T$ to be equivariant using equivariant neural networks.

For training these networks, we employ the same self-supervised contrastive loss in Kipf et al. (2020). Let $(s, a, s')$ be a ground truth transition triplet and $s''$ be an incorrect next state $s'' \neq s'$. Let $z = E(S(s))$, $z' = E(S(s))$, and $z'' = E(S(s))$, then

$$\mathcal{L}(s, a, s', s'') = \|T(z, a) - z'\| + \alpha \max(\beta - \|T(z, a) - E(z'')\|, 0).$$

Minimizing this loss pushes $T(z, a)$ towards $z'$ and away from the incorrect sample $z''$.

**Symmetric Embedding Network $S$**    We use non-equivariant CNNs for all environments, but the specific architecture varies (see Appendix D.2). For example, in the object-centric environments, the symmetry group is $G = C_4 \times S_5$, the cyclic group of order 4 acting by $\pi/2$-rotations and the permutation group on the 5 objects. The output $y$ has shape $[B, C, 5, 4]$ and carries an action $\rho_{\mathcal{Y}}$ of $S_5$ by permuting the 5-dimensional axis and of $C_4$ by cyclically permuting the 4-dimensional axis.

**Equivariant Encoder $E$ and Transition Model $T$**    The encoder and transition model are both implemented using $G$-equivariant neural networks. In the object-centric environments with $G = C_4 \times S_5$, the equivariant encoder $E$ is shared over all 5 objects and uses group convolution over $C_4$ (Cohen & Welling, 2016a), thus achieving $C_4 \times S_5$-equivariance. The transition function $T$ is implemented as a GNN (message-passing neural network) with edge and node networks which use $C_4$-convolutions for their linear layers. Since GNNs are $S_5$ equivariant by definition and the linear layers within the GNNs are $C_4$-equivariant, this is $C_4 \times S_5$-equivariant. For other implementations, see Table 1 and Appendix D.2.

## 4.2 SO(3)-Structured Symmetric Embeddings

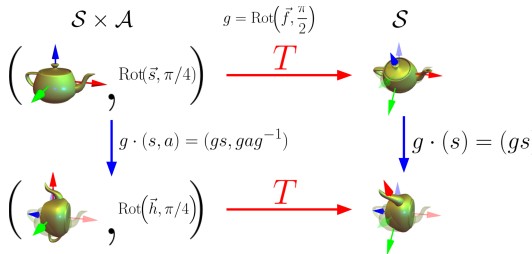

Figure 3: SO(3)-equivariance of the transition function for SO(3) object manipulation.

For these environments, we consider a case where the action space $\mathcal{A}$ is the same as the symmetry group $G = \mathcal{A} = \mathrm{SO}(3)$, similar to Quessard et al. (2020). We choose to model the latent space $\mathcal{Z}$ using SO(3) as well, which is not a linear group representation, but a set with a group action.

**Symmetric Embedding** In the case of $G = \mathrm{SO}(3)$ symmetry, we expect the symmetric embedding network to detect the pose of object $z$ in 3D. We omit the equivariant encoder (by setting $E = \mathrm{id}$) and instead use a two-part Symmetric Embedding Network that directly encodes $y = z$ using a down-sampling CNN whose output is passed to an MLP, and converted to an element of SO(3).

To force the output of the symmetric embedding network $y$ to be an element of SO(3), we have the last layer output 2 vectors $u, v \in \mathbb{R}^3$ and perform Gram-Schmidt orthogonalization to construct a positively oriented orthonormal frame (see Appendix). This method is also used by Falorsi et al. (2018), who conclude it produces less topological distortion than alternatives.

**Transition** We implement an equivariant transition model using Tensor Field Networks (Thomas et al., 2018; Geiger et al., 2020). This is an $\mathrm{SO}(3) \ltimes (\mathbb{R}^3, +)$-equivariant method which works over point clouds. Here $z \in \mathrm{SO}(3)$ and $a \in \mathrm{SO}(3)$. We consider $z$ as 3 points in $\mathbb{R}^3$ and add the origin to get a 4 point cloud. We embed the actions as features over these 4-points. The MDP action $a$ is then set as a feature over these 4 points, which has SO(3)-representation $\rho_{\mathcal{A}}(g) \cdot a = gag^{-1}$.

The MDP action $a \in \mathrm{SO}(3)$ is a rotation matrix and the latent state $z \in \mathrm{SO}(3)$ is a positively-oriented orthogonal coordinate frame. Though $\mathcal{Z} = \mathcal{A} = \mathrm{SO}(3)$, these different semantics lead to differing $G = \mathrm{SO}(3)$ actions with $\rho_{\mathcal{Z}}(g)(z) = g \cdot z$ but $\rho_{\mathcal{Z}}(g)(a) = gag^{-1}$. If $z$ is correctly learned, then the ground truth latent transition function can be represented by a simple matrix multiplication $T_{\mathcal{Z}}(z, a) = az$ which is also equivariant,

$$T_{\mathcal{Z}}(\rho_{\mathcal{Z}}(g)(z), \rho_{\mathcal{A}}(g)(a)) = (gag^{-1})(gz) = gaz = \rho_{\mathcal{Z}}(g)T_{\mathcal{Z}}(z, a).$$

This method, which we label `MatMul`, is similar to the latent transition model used in Quessard et al. (2020), except in our framework the groundtruth $a \in \mathrm{SO}(3)$ is provided to aid learning $z$.

## 4.3 Generalizing over the MDP Action Space

Although the state does not have a known group action $\rho_{\mathcal{S}}$, the MDP action does have known $\rho_{\mathcal{A}}$. In the domains we consider, although the state is high-dimensional and has non explicit symmetry, the action is low dimensional and has clear symmetry. The MDP action is passed directly to $T$ and thus bypasses the non-equivariant part $S$ of the neural network. Since the neural network is explicitly equivariant with respect to the MDP action, it is thus feasible to train the neural network using only a proper subset $\mathcal{A}' \subset \mathcal{A}$ of the action space, and then test on the entire $\mathcal{A}$. This may be useful in domains in which data collection is costly. Since the samples from $\mathcal{S}$ are still i.i.d., the non-equivariant neural network $S$ is still able to learn well.

In generalization experiments, we require that $\rho_{\mathcal{A}}(G) \cdot \mathcal{A}' = \mathcal{A}$, i.e. every MDP action is $G$-transformed version of one in $\mathcal{A}'$. We assume that $S$ is approximately equivariant after training, which, as $T$ and $E$ are constrained to be equivariant, implies $T_{\mathcal{S}}(s, a) = T(E(S(s)), a)$ is equivariant. We also assume low error for $T$ on the restricted action set $(s_1, a', s_2) \in \mathcal{S} \times \mathcal{A}' \times \mathcal{S}$. Then given $(s_1, a, s_2) \in \mathcal{S} \times \mathcal{A} \times \mathcal{S}$, there exists $g \in G$ such that $a = \rho_{\mathcal{A}}(g) \cdot a'$. Let $s'_i = \rho_{\mathcal{S}}(g^{-1})s_i$. Then

$$T_{\mathcal{S}}(s, a) \approx T_{\mathcal{S}}(\rho_{\mathcal{S}}(g) \cdot s'_1, \rho_{\mathcal{A}}(g) \cdot a') \approx \rho_{\mathcal{S}}(g) \cdot T_{\mathcal{S}}(s'_1, a') \approx \rho_{\mathcal{S}}(g) \cdot s'_2 \approx s_2.$$

If performance is good for $\mathcal{A}'$ and $S$ has low equivariance error, then performance will be good for $\mathcal{A}$.

## 5 EXPERIMENTS

### 5.1 SETUP

We choose five environments with varying symmetries to evaluate our models. The first three environments 2D Shapes, 3D Blocks, and Rush Hour are grid-worlds with five moving objects, based on (Kipf et al., 2020). Rush Hour is a variant of 2D Shapes where objects can move relative to their orientation. We consider symmetry to $\pi/2$ rotations and object permutations. We also evaluate in a continuous control domain, the `Reacher-v2` MuJoCo environment, with rotational and translational symmetries. The last domain is of a rotating 3D teapot, where an action is an element of $SO(3)$. We consider two action spaces: a small discrete action space (S) with 6 rotations of $\frac{2\pi}{30}$ in $SO(3)$ and a large continuous action space (L) of any rotation in $SO(3)$. All environments use images as observed states. Additional details are given in the Appendix D.1.

We compare three types of models: (a) a non-equivariant model with no enforced symmetry, (b) a fully-equivariant model with a mis-specified symmetry where $\rho_{\mathcal{S}}$ is a simple transformation of the pixels, and (c) our method. For 3D Teapot, we forgo the fully equivariant baseline as it is hard to define a $\rho_{\mathcal{S}}$ acting on the 2D image space which approximates the true group action.

### 5.2 MODEL ARCHITECTURES AND TRAINING

As each environment contains different symmetries, the model architecture is customized for every environment keeping the meta-architecture the same. We use object-oriented structured models which factorize the latent state space and latent action space over objects for the grid world environments of 2D shapes, 3D blocks, and Rush Hour. Though the objects and actions are factorized, the world model must account for the pairwise interactions between objects (e.g. actions to move one object can be blocked by another object). The encoder $E$ is shared over all objects and a GNN is used for the transition function $T$. The Reacher and 3D Teapot environments do not consider objects and thus we do not use model components that consider permutations. A summary of the environments, different symmetries, representation types, and model architectures are given in Table 1.

We do not consider the reward as our focus is on constructing accurate latent representations and their dynamics. A random policy was used to create training and evaluation datasets of $(s, a, s')$ tuples. For all environments, we have either a combinatorially large state space (with objects) or continuous states and thus overlap between training and evaluation datasets is highly unlikely.

As equivariant networks have more parameters than the non-equivariant counterparts, we reduce the number of hidden dimensions accordingly to keep the number of parameters approximately constant for all models. The Adam (Kingma & Ba, 2014) optimizer was used for all experiments. All other specific implementation details are provided in Appendix D.2.

### 5.3 METRICS

In order to evaluate the performance of our model in latent space without reconstruction, we use standard ranking metrics from Kipf et al. (2020), modified versions of these metrics to adapt to continuous state spaces, and two metrics for evaluating the equivariance of the learned model.

#### 5.3.1 ACCURACY METRICS

**Hits, Hard Hits, and MRR**  The evaluation samples are ranked according to the pairwise $L_2$ distance of the predicted next states and the true next states (both are encoded in latent space). Hits at Rank $k$ (H@k) measures the average percentage of time that the predicted next state is within $k$-nearest neighbors of the encoded true next state. The mean reciprocal rank (MRR) is the average inverse rank. We also consider a variant of Hits at Rank $k$ (HH@k) where we generate negative samples $s'_n$ of states that are close to the true next state $s'$ (see Appendix for more details) and count the number of times that the distance to the positive sample was lower than the distance t o the negative samples. This is a harder version of H@k as the model must distinguish between close negative samples and the true positive sample in latent space.

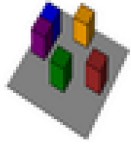 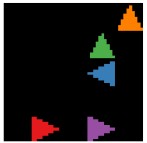 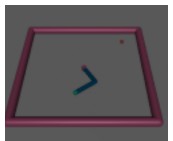 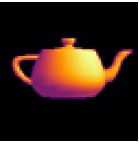

| Environment | 2D Shapes & 3D Blocks | Rush Hour | Reacher | 3D Teapot |
|---|---|---|---|---|
| Observation $s$ | 50x50x3 | 50x50x3 | 128x128x3x2 | 64x64x1 |
| Action $a$ | { up,right,down,left } | { fwd,left,back,right } | $(\phi_1'', \phi_2'') \in \mathbb{R}^2$ (joint forces) | SO(3) |
| Symmetry $G$ | $C_4 \times S_5$ ($\frac{\pi}{2}$ rot.; obj. perm.) | $C_4 \times S_5$ ($\frac{\pi}{2}$ rot.; obj. perm.) | $D_4 \ltimes (\mathbb{R}^2, +)$ ($\frac{\pi}{2}$ rot; flip; trans.) | SO(3) |
| $\mathcal{Z}$-rep: $\rho_{\mathcal{Z}}$ | $(\rho_{\text{std}}, \mathbb{R}^2) \boxtimes (\rho_{\text{std}}, \mathbb{R}^5)$ | $(\rho_{\text{std}} \oplus \rho_{\text{reg}}, \mathbb{R}^6)$ $\boxtimes (\rho_{\text{std}}, \mathbb{R}^5)$ | $(\rho_{\text{reg}}, \mathbb{R}^8)^4 \boxtimes \rho_{\text{triv}}$ | $gz$ (matrix mult.) |
| $\mathcal{A}$-rep: $\rho_{\mathcal{A}}$ | $(\rho_{\text{reg}}, \mathbb{R}^4) \boxtimes (\rho_{\text{std}}, \mathbb{R}^5)$ | $(\rho_{\text{triv}}, \mathbb{R})^4$ | $(\rho_{\text{flip}}, \mathbb{R})^2$ | $gag^{-1}$ (conjugation) |
| Non-Equ. Extractor | 2-layer CNN (2D) 4-layer CNN (3D) | 2-layer CNN | 7-layer CNN | 4 conv, 3 FC layers |
| Equ. Encoder | MLP + $C_4$-conv | MLP + $C_4$-conv | 3 $E(2)$-conv, 3 $D_4$-FC layers | Id. (None) |
| Equ. Transition | GNN + $C_4$-conv Cohen & Welling (2016a) Scarselli et al. (2008) | GNN + $C_4$-conv | MLP + $E(2)$-CNN Weiler & Cesa (2019) | MLP + Tensor Field Geiger et al. (2020) *or* matrix mult. |

Table 1: The symmetry and implementation for each domain. See Appendix E for the $\rho$ definitions.

### 5.3.2 EQUIVARIANCE METRICS

**Equivariance Error** EE    In order to analyze the equivariance of each model, we generate a version of the evaluation dataset where one element of the symmetry group acts on the tuple $(s, a, s')$ and calculate the true equivariance error for the embedding network. Although by assumption $\rho_{\mathcal{S}}(g) \cdot s$ cannot be computed using $g$ and $s$, our synthetic datasets allow us to render both $s$ and $\rho_{\mathcal{S}}(g) \cdot s$ during generation. Specifically, the equivariance error of the symmetric embedding is calculated as

$$\text{EE} = \mathbb{E}_{s,g} \left[ \| \rho_{\mathcal{Y}}(g) \cdot S(s) - S(\rho_{\mathcal{S}}(g) \cdot s) \| \right].$$

**Distance Invariance Error** DIE    The above equivariance error can always be applied to the symmetric embedding network when its output space is spatial and we can manually perform group actions on the outputs. However it cannot be applied to the latent space $\mathcal{Z}$ in the case of non-equivariant models since the group action on the latent space $\rho_{\mathcal{Z}}$ cannot be meaningfully defined.

We therefore propose a proxy for the equivariance error using invariant distances. For a pair of input states $s, s'$, an equivariant model $f$ will have the same distances $\|f(s) - f(s')\|$ and $\|f(gs) - f(gs')\|$ assuming the action of $G$ is norm preserving as it is for all transformations considered in the paper. Due to the linearity of the action, $\|f(gs) - f(gs')\| = \|gf(s) - gf(s')\| = \|g(f(s) - f(s'))\| = \|(f(s) - f(s'))\|$. The distance invariance error is computed as

$$\text{DIE} = \mathbb{E}_{s,s',g} \left[ | \|f(s) - f(s')\| - \|f(gs) - f(gs')\| | \right].$$

We evaluate both the symmetric embedding network (DIE(S)) and entire model (DIE(model)).

### 5.4 MODEL PERFORMANCE COMPARISON

The results are shown in Tables 2,3. In general, the ranking metrics (Hits and MRR) show that all three models are accurate on the 3D blocks, Rush Hour, and Reacher environments. The non-equivariant model achieves a higher H@1 on Rush Hour but has a slightly lower MRR than either the fully equivariant and our model on Reacher. Surprisingly, the fully equivariant model performs very well even when the group action on the input space $\rho_{\mathcal{S}}$ is not correct. Due to the skewed perspective, we can see that the simple pixel-level transformation maps training data to out-of-distribution images which are never seen by the model. We hypothesize that equivariance does not hamper its performance

| | Model | Hits@1 (10 step, %) | MRR (10 step, %) | EE($S$) | DIE($S$) ($\times 10^{-2}$) | DIE (model) (10 step, $\times 10^{-2}$) |
|---|---|---|---|---|---|---|
| 3D Blocks | None | $94.3_{\pm 9.0}$ | $99.0_{\pm 1.5}$ | $0.89_{\pm 0.3}$ | $3.30_{\pm 1.6}$ | $3.85_{\pm 2.0}$ |
| | Full | $99.8_{\pm 0.3}$ | $99.9_{\pm 0.2}$ | $0.82_{\pm 0.5}$ | $3.36_{\pm 2.2}$ | $5.54_{\pm 4.8}$ |
| | Ours | $99.9_{\pm 0.0}$ | $100_{\pm 0.0}$ | $0.86_{\pm 0.4}$ | $3.32_{\pm 1.9}$ | $3.16_{\pm 1.5}$ |
| Rush Hour | None | $95.9_{\pm 1.3}$ | $97.9_{\pm 0.7}$ | $0.37_{\pm 0.07}$ | $5.17_{\pm 1.31}$ | $26.6_{\pm 7.13}$ |
| | Full | $92.0_{\pm 3.1}$ | $95.9_{\pm 1.7}$ | $0.00_{\pm 0.00}$ | $0.00_{\pm 0.00}$ | $0.05_{\pm 0.06}$ |
| | Ours | $93.3_{\pm 3.7}$ | $96.5_{\pm 2.0}$ | $0.26_{\pm 0.09}$ | $4.12_{\pm 1.77}$ | $10.0_{\pm 3.45}$ |

Table 2: Model Performance on 3D Blocks and Rush Hour.

| | Model | H@10 (1 step, %) | MRR (1 step, %) | EE($S$) | DIE($S$) ($\times 10^{-2}$) | DIE (model) (1 step) |
|---|---|---|---|---|---|---|
| Reacher | None | $100_{\pm 0.0}$ | $88.3_{\pm 3.3}$ | $1.26_{\pm 0.1}$ | $4.53_{\pm 1.1}$ | $0.56_{\pm 0.2}$ |
| | Full | $100_{\pm 0.0}$ | $95.5_{\pm 1.9}$ | $1.19_{\pm 0.0}$ | $3.51_{\pm 0.7}$ | $0.39_{\pm 0.1}$ |
| | Ours | $100_{\pm 0.0}$ | $94.1_{\pm 2.8}$ | $1.29_{\pm 0.0}$ | $4.05_{\pm 0.7}$ | $0.52_{\pm 0.1}$ |

| | HH@1 (1 step, %) | EE($S$) |
|---|---|---|
| None (S) | $0.1_{\pm 0.1}$ | $2.34_{\pm 0.1}$ |
| MatMul (S) | $31.6_{\pm 1.2}$ | $2.39_{\pm 0.0}$ |
| TFN (S) | $39.7_{\pm 1.2}$ | $2.22_{\pm 0.1}$ |
| None (L) | $7.4_{\pm 1.4}$ | $2.409_{\pm 0.0}$ |
| MatMul (L) | $100_{\pm 0.0}$ | $0.05_{\pm 0.0}$ |
| TFN (L) | $4.9_{\pm 0.7}$ | $2.41_{\pm 0.0}$ |

Reacher        3D Teapot

Table 3: Model performance on Reacher (left) and 3D Teapot (right) environments. For the 3D Teapot models, None is the non-equivariant model, Matmul is the matrix multiplication model and Equiv is the version using Tensor Field Networks. (S) denotes the small discrete action space with 6 rotations of $\frac{2\pi}{30}$ in $SO(3)$ and (L) denotes any rotation in $SO(3)$.

on training data, but only constrains its extrapolation capabilities to out-of-distribution samples. In 3D Teapot, we observe that our Tensor Field Networks (TFN) and Matrix Multiplication (MatMul) models outperform the non-equivariant model, but have different performance for different action spaces. With small actions (S), Tensor Field Networks (TFN) and Matrix Multiplication (MatMul) perform similarly, while TFN fails with large actions (L). We hypothesize that the difference is caused by the contrastive loss function; small actions can lead to local minima where all states get mapped to close-by latent states. For the equivariance metrics, our model outperforms the non-equivariant model in all domains on DIE(model), while it performs similarly on EE(S) and DIE(S). As was the case for accuracy metrics, the fully equivariant model performs surprisingly well with possibly the same reason outlined above.

**Visualization of latent embeddings** We visualize the latent embedding $z$ for our model to qualitatively analyze what kind of representations are learned. All figures are provided in the Appendix for space. Figure 4 plots all the learned embeddings for Reacher for all observations in the evaluation set and shows a sample transition in both pixel and latent space. The encoded current state $z$ is highlighted in red and the encoded next state is highlighted $z'$ is highlighted in orange. which we factor into irreducible representations (irreps) before visualizing (see Hall (2003)). Some irreducible representations are 1-dimensional and are plotted as a line. The 2-dimensional irreps show a clear circular pattern, match the joint rotations of the environment.

Figure 5 shows the traversal of rotations in pixel and latent space for 3D Teapot. The latent space can choose its own base coordinate frame and thus is oriented downwards. We can clearly see that the effective rotations relative to the objects' orientation perfectly align, demonstrating that the learned embeddings correctly encode 3D poses and rotations.

## 5.5 GENERALIZATION FROM LIMITED ACTIONS

We now train on a limited subset of actions and evaluate on datasets generated with the full action space. These experiments aim to verify that our model, even where all components are not designed to be equivariant, can learn a good equivariant representation which can generalize to unseen actions.

We perform experiments on the 2D Shapes, 3D Blocks, and Reacher domains. For 2D Shapes, the training data only contains 'up' actions and we set the number of episodes to 100,000 with length 1 to avoid any distribution shifts in the data (e.g. performing up continuously will produce many transitions where all blocks are blocked by the boundaries). For 3D Blocks, we omit the left action in training and use similar modifications to the episode lengths. For Reacher, the action space represents

| | H@1 (10 step, %) | MRR (1 step, %) | EE($S$) | DIE($S$) ($\times 10^4$) | DIE (model) (10 step) |
|---|---|---|---|---|---|
| CNN | $2.8_{\pm 0.6}$ | $5.3_{\pm 0.4}$ | $0_{\pm 0}$ | $0_{\pm 0}$ | $0.19_{\pm 0.05}$ |
| Ours/Full | $100_{\pm 0.0}$ | $99.9_{\pm 0.0}$ | $0_{\pm 0}$ | $0_{\pm 0}$ | $0_{\pm 0}$ |

Table 4: Generalization results for 2D Shapes trained using only the up action and evaluated on all actions. Due to the simplicity of the environment, a simple CNN turns out to be equivariant, so both the baseline CNN and Ours/Full have an equivariant symmetric embedding network.

| | H@1 (10 step, %) | MRR (10 step, %) | EE($S$) | DIE($S$) ($\times 10^{-2}$) | DIE (model) (10 step, $\times 10^{-3}$) |
|---|---|---|---|---|---|
| None | $52.3_{\pm 14}$ | $61.8_{\pm 13}$ | $0.98_{\pm 0.2}$ | $3.64_{\pm 1.5}$ | $181_{\pm 79}$ |
| Full | $83.7_{\pm 36}$ | $86.0_{\pm 31}$ | $0.81_{\pm 0.5}$ | $3.32_{\pm 2.1}$ | $14.8_{\pm 9.1}$ |
| Ours | $99.9_{\pm 0.0}$ | $100_{\pm 0.0}$ | $0.96_{\pm 0.3}$ | $3.65_{\pm 1.6}$ | $5.0_{\pm 4.7}$ |

Table 5: Generalization results for 3D Blocks with limited actions. The training set only contains the up, right and down actions; the evaluation set contains all four actions.

| | H@10 (1 step, %) | MRR (1 step, %) | EE($S$) | DIE($S$) ($\times 10^{-2}$) | DIE (model) (1 step, $\times 10^{-2}$) |
|---|---|---|---|---|---|
| None | $86.5_{\pm 3.0}$ | $50.6_{\pm 3.1}$ | $1.22_{\pm 0.1}$ | $6.54_{\pm 1.9}$ | $6.95_{\pm 1.5}$ |
| Full | $89.4_{\pm 11}$ | $61.8_{\pm 13}$ | $1.18_{\pm 0.1}$ | $3.62_{\pm 0.8}$ | $4.87_{\pm 1.4}$ |
| Ours | $90.8_{\pm 4.5}$ | $59.4_{\pm 4.6}$ | $1.28_{\pm 0.1}$ | $5.45_{\pm 0.9}$ | $4.95_{\pm 0.6}$ |

Table 6: Reacher with limited actions. The models were trained on data where the second joint is constrained to be positive and evaluated on unconstrained data.

joint actuation forces $\in [-1, 1]$ for each of the two joints. We restrict the range of the force for the second joint to be positive, meaning that the second arm rotates in only one direction.

Tables 4,5,6 show results for 2D Shapes, 3D Blocks, and Reacher respectively. We see that our method can successfully generalize over unseen actions compared to both the non-equivariant and fully equivariant baselines. The non-equivariant baseline in particular performs poorly on all domains, achieving only 2.8% on Hits@1 and 5.5% on MRR for 2D Shapes. The fully equivariant model performs worse than our method for 3D Blocks and achieves a similar performance on Reacher. As the fully equivariant model performs well when trained on all actions but does not perform as well in these generalization experiments, these results lend support to our hypothesis that the incorrect pixel-level equivariance bias limits its extrapolation abilities to out-of-distribution samples. In these limited actions experiments, the fully equivariant model cannot extrapolate correctly and achieves lower performance than our model.

Figure 6 shows embeddings for all states in the evaluation dataset for our model and the non-equivariant model trained on only the up action. Our model shows a clear $5 \times 5$ grid, while the non-equivariant model learns a degenerate solution (possibly encoding only the row index $x$).

## 6 CONCLUSION AND FUTURE WORK

We demonstrate a flexible method which can be used to extend equivariant neural networks to domains with known symmetry types, but transformation properties which cannot be easily explicitly described. We apply our method across a variety of domains and equivariant neural network architectures. Our methods confer some of the advantages of equivariant neural networks in situations where they did not previously apply, such as generalization to data outside the training distribution. Future work will include applying our method to tasks besides world models and using our method to develop disentangled and more interpretable features in domains with known but difficult to isolate symmetry.

ETHICS STATEMENT

Our paper does directly address domains with privacy or safety concerns. However, our method can be used in robotics applications to train robots using fewer data samples. To the extent that robotics technology can be used for benefit or harm, our method enables both options.

REPRODUCIBILITY STATEMENT

We will open-source our code, including all models and data generation scripts, thus allowing all experiments to be fully reproduced.

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

# A OUTLINE

Our appendix is organized as follows. First, in Section B, we provide an additional formal setup of the problem. The visualization of learned latent symmetric representations is presented in Section C, followed by the details of training environments and network architectures in Section D.1 and D.2. We further explain the notation and definition in Section E.

# B SETUP: EQUIVARIANT WORLD MODELS

In this section, we provide a technical background for building equivariant world models, which we use in learning symmetric representations.

We model our interactive environments as Markov decision processes. A (deterministic) Markov decision process (MDP) is a 5-tuple $\mathcal{M} = \langle \mathcal{S}, \mathcal{A}, T, R, \gamma \rangle$, with state space $\mathcal{S}$, action space $\mathcal{A}$, (deterministic) transition function $T : \mathcal{S} \times \mathcal{A} \to \mathcal{S}$, reward function $R : \mathcal{S} \times \mathcal{A} \to \mathbb{R}$, and discount factor $\gamma \in [0, 1]$.

Symmetry can appear in MDPs naturally (Zinkevich & Balch, 2001; Narayanamurthy & Ravindran, 2008; Ravindran, 2004; van der Pol et al., 2020b), which we can exploit using equivariant networks. For example, van der Pol et al. (2020b) study geometric transformations, such as reflections and rotations. Ravindran (2004) study group symmetry in MDPs as a special case of MDP homomorphisms.

**Symmetry in MDPs.** Symmetry in MDPs is defined by the automorphism group $\mathrm{Aut}(\mathcal{M})$ of an MDP, where an automorphism $g \in \mathrm{Aut}(\mathcal{M})$ is an MDP homomorphism $h : \mathcal{M} \to \mathcal{M}$ that maps $\mathcal{M}$ to itself and thus preserves its structure. Zinkevich & Balch (2001) show the invariance of value function for an MDP with symmetry. Narayanamurthy & Ravindran (2008) prove that finding exact symmetry in MDPs is graph isomorphism complete.

Ravindran (2004) provide a comprehensive overview of using MDP homomorphisms for state abstraction and study symmetry in MDPs as a special case. A more recent work by van der Pol et al. (2020b) builds upon the notion of MDP homomorhpism induced by group symmetry and uses it in an inverse way. They assume knowledge of MDP homomorphism induced by symmetry group is known and exploit it. Different from us, their focus is on policy learning, which needs to preserve both transition and reward structure and thus has *optimal value equivalence* (Ravindran, 2004).

More formally, an *MDP homomorphism* $h : \mathcal{M} \to \overline{\mathcal{M}}$ is a mapping from one MDP $\mathcal{M} = \langle \mathcal{S}, \mathcal{A}, T, R, \gamma \rangle$ to another $\overline{\mathcal{M}} = \langle \overline{\mathcal{S}}, \overline{\mathcal{A}}, \overline{T}, \overline{R}, \gamma \rangle$ which needs to preserve the transition and reward structure (Ravindran, 2004). The mapping $h$ consists of a tuple of surjective maps $h = \langle \phi, \{\alpha_s \mid s \in \mathcal{S}\} \rangle$, where $\phi : \mathcal{S} \to \overline{\mathcal{S}}$ is the state mapping and $\alpha_s : \mathcal{A} \to \overline{\mathcal{A}}$ is the *state-dependent* action mapping. The mappings are constructed to satisfy the following conditions: (1) the transition function is preserved $\overline{T}\left(\phi\left(s'\right) \mid \phi(s), \alpha_s(a)\right) = \sum_{s'' \in \phi^{-1}(\phi(s'))} T\left(s'' \mid s, a\right)$, (2) and the reward function is also preserved $\overline{R}\left(\phi(s), \alpha_s(a)\right) = R(s, a)$, for all $s, s' \in \mathcal{S}$ and for all $a \in \mathcal{A}$.

An MDP isomorphism from an MDP $\mathcal{M}$ to itself is call an *automorphism* of $\mathcal{M}$. The collection of all automorphisms of $\mathcal{M}$ along with the composition of homomorphisms is the *automorphism group* of $\mathcal{M}$, denoted $\mathrm{Aut}(\mathcal{M})$.

We specifically care about a subgroup of $G \subseteq \mathrm{Aut}(\mathcal{M})$ which is usually easily identifiable from environments a priori and thus we can design appropriate equivariant network architectures to respect it, such as $C_4$ rotation symmetry of objects. Additionally, while MDP homomorphisms pose constraints to the transition and reward function, we only care about the transition function, especially the *deterministic* case $T : \mathcal{S} \times \mathcal{A} \to \mathcal{S}$.

**Equivariant transition.** By definition, when an MDP $\mathcal{M}$ has symmetry, any state-action pair $(s, a)$ and its transformed counterpart $(\rho_{\mathcal{S}}(g) \cdot s, \rho_{\mathcal{A}}(g) \cdot a)$ are mapped to the same abstract state-action pair by $h \in Aut(\mathcal{M})$: $(\phi(s), \alpha_s(a)) = (\phi(gs), \alpha_{gs}(ga))$, for all $s \in \mathcal{S}, a \in \mathcal{A}, g \in G$. Therefore, the transition function $T : \mathcal{S} \times \mathcal{A} \to \mathcal{S}$ should be $G$-equivariant:

$$T(\rho_{\mathcal{S}}(g) \cdot s, \rho_{\mathcal{A}}(g) \cdot a) = \rho_{\mathcal{S}}(g) \cdot T(s, a), \qquad (1)$$

for all $s \in \mathcal{S}, a \in \mathcal{A}, g \in G$.

**State-dependent action transformation.** Note that the group operation acting on action space $\mathcal{A}$ *depends on state*, since $G$ actually acts on the *product space* $\mathcal{S} \times \mathcal{A}$: $(g, (s, a)) \mapsto \rho_{\mathcal{S} \times \mathcal{A}}(g) \cdot (s, a)$. However, in most cases, including all of our environments, the action transformation $\rho_{\mathcal{A}}$ does not depend on state. As a bibliographical note, the formulation in van der Pol et al. (2020b) also has a joint group action on state and action space, which is denoted as state transformation $L_g : \mathcal{S} \rightarrow \mathcal{S}$ and *state-dependent* action transformation $K_g^s : \mathcal{A} \rightarrow \mathcal{A}$. Table 1 in van der Pol et al. (2020b) outlines state and action transformations for their environments, and all of action transformations are not state-dependent.

Similarly in our case, geometric transformations are usually acting globally on the environments $\mathcal{S} \times \mathcal{A}$, thus states and actions are transformed accordingly. We use the factorized form and omit the state-dependency $\rho_{\mathcal{A}}(g; s)$ of action transformation $\rho_{\mathcal{A}}(g)$, since the action transformations do not depend on states $\rho_{\mathcal{A}}(g; s) = \rho_{\mathcal{A}}(g)$ for all $g \in G$, $s \in \mathcal{S}$.

**Learning transition with equivariant networks.** In this paper we are mainly interested in learning transition functions which are equivariant under symmetry transformations and can be high-dimensional.

We apply the idea of learning equivariant transition models in the latent space $\mathcal{Z}$, where $\mathcal{Z}$ is the space of symmetric representations, on various environments with different symmetry groups $G$. We assume we do not explicitly know $\rho_{\mathcal{S}}$ since $\mathcal{S}$ is high-dimensional. We factorize the group representation on state and action $\mathcal{S} \times \mathcal{A}$ as latent state transformation $\rho_{\mathcal{Z}}(g) \cdot E(s)$ and $\rho_{\mathcal{A}}(g; s) \cdot a$. In the deterministic case, the transition model can be modeled by $G$-equivariant networks in latent state $\mathcal{Z}$ and action space $\mathcal{A}$:

$$\rho_{\mathcal{Z}}(g) \cdot T(E(s), a) = T(\rho_{\mathcal{Z}}(g) \cdot E(s), \rho_{\mathcal{A}}(g) \cdot a), \qquad (2)$$

for all $g \in G$, $s, s' \in \mathcal{S}$ and $a \in \mathcal{A}$.

## C  LEARNED LATENT REPRESENTATIONS

The learned latent embedding $z$ for all states in the evaluation set for Reacher is shown in Figure 4 and the embeddings are factored into irreducible representations. The 2-dimensional representations show a circular pattern, mimicking the rotation of joints. Figure 5 shows a sample observation in pixel space (ground truth) in the top row and its encoded latent embedding in the bottom row. The latent space can have a different orientation than the ground truth. Applying rotations to the low-dimensional embedding yields a latent space traversal with smooth rotations, showing that the learned representations correctly encode the correct symmetries. Figure 6 show learned embeddings for all states in the evaluation dataset for our model (left) and the non-equivariant model (right) when trained on only the up action.

## D  TRAINING DETAILS

### D.1  ENVIRONMENTS

**2D Shapes & 3D Blocks** There are five objects are arranged in a $5 \times 5$ grid and each object can occupy a single cell. Actions are the 4 cardinal directions for each object and an action moves one object at a time, unless it is blocked by the boundaries or by another object. Observations are $50 \times 50 \times 3$ RGB images for both 2D Shapes and 3D Blocks, with pixel values normalized to $[0, 1]$. The observations in 2D shapes are top down views of the grid and each object has a different color-shape combination. For 3D Blocks, the observations are rendered isometrically with a skewed perspective and each block has a $z$-height, introducing partial occlusions to the image.

**Rush Hour** We create a variant of 2D Shapes called Rush Hour. Each object has an orientation and the action is relative to the object's orientation: {forward, backward, left, right}. This increases the importance of rotational orientation in the environment increasing the significance of equivariance.

**Reacher** This environment makes a small modification to the original MuJoCo environment `Reacher-v2`. As we do not consider rewards, we fix the goal position to the position $[0.2, 0.2]$ so

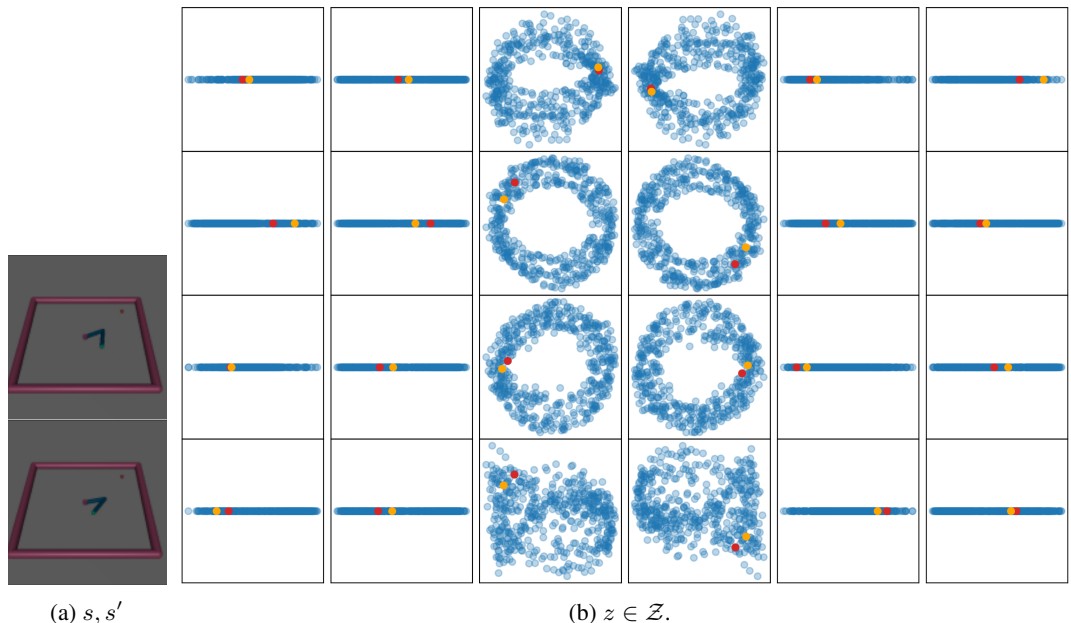

(a) $s, s'$          (b) $z \in \mathcal{Z}$.

Figure 4: Learned symmetric embeddings for Reacher: pixel observation $s$ (left top) and next observation $s'$ (left bottom), latent representation $z$ of the evaluation set (right). The representation type of $z$ is $\rho_{D_4,\text{reg}}$ which we factor into irreducible representations before visualizing (see Hall (2003)). All encoded samples in the evaluation set are shown and the encoded current observation is colored red and the encoded next observation is colored orange. There is a clear circular pattern that match joint rotations.

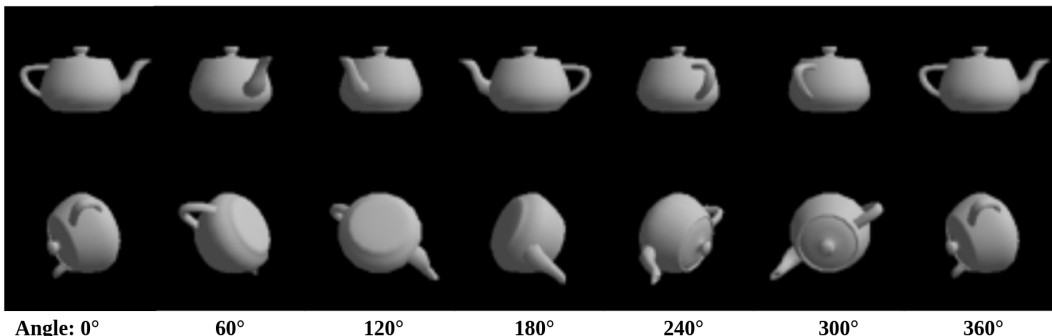

Angle: 0°     60°     120°     180°     240°     300°     360°

Figure 5: Latent space traversal for matrix multiplication (Matmul) transition model in Teapot. Top row: Ground truth rotation of a teapot. Bottom row: Projection in the latent space of the Matmul model, which is regularized to consist only of valid rotation matrices. The model, by construction, does not have a fixed reference frame; hence, the two sequences are offset by the learned latent reference frame.

that features related to the goal are ignored. Instead of using the 11-dimensional state, we use pixel observations as images and preprocess them by cropping slightly and downsampling the original $500 \times 500 \times 3$ RGB image to $128 \times 128 \times 3$. The previous and current frames are then stacked as an observation to encode velocities. The default camera position gives a slightly skewed perspective, see Table 1.

**3D Teapot** The 3D teapot enviroment contains images of the Utah teapot model rendered into the $64 \times 64$ grayscale images. The teapot varies in pose which can be described by a coordinate frame $z \in \text{SO}(3)$. We consider both a small (S) and large (L) action space for this environment. In the small action space, 6 actions may be taken corresponding to multiplication of the pose $z \mapsto az$ by

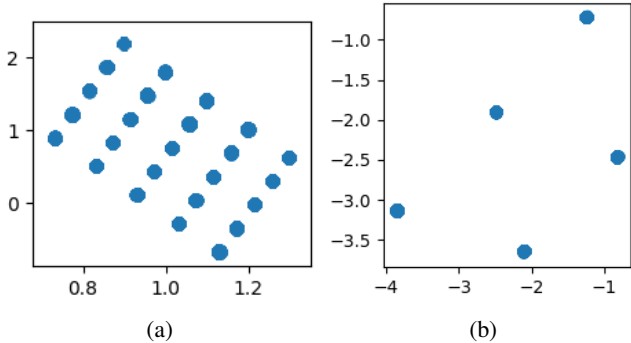

(a)                                    (b)

Figure 6: 2D Shapes: learned embeddings for all states in the evaluation set when trained on only the up action. Our model (left) is able to generalize well and learns the correct underlying $5 \times 5$ grid. The non-equivariant model (right) learns a degenerate solution

$a \in \mathrm{SO}(3)$ where $a$ is a rotation of $\pi 2\pi/30$ around the $x-$,$y-$,or $z$-axis. In the large action space, actions may be any element $a \in \mathrm{SO}(3)$.

## D.2 MODEL ARCHITECTURES

**Symmetric embedding network** $S$    For all models and environments except for 3D Teapot, we use CNNs with BatchNorm Ioffe & Szegedy (2015) and ReLU activations between each convolutional layer. For 3D Teapot, the symmetric embedding network maps directly to the latent $z$ space so we use 4 convolutional layers followed by 3 fully connected layers. The output is a $3 \times 3$ rotation matrix. The number of layers for each environment is given in Table 1. For the non-equivariant symmetric embedding networks, we use 32 convolutional filters for every layer and use 8 filters for Reacher and 16 filters for all other environments.

**Encoder** $E$    The object-oriented environments use 3-layer MLPs with 512 hidden units for the non-equivariant networks and 256 for the equivariant counterparts. There is a ReLU activation after the first and second layers and a LayerNorm (Ba et al., 2016) after the second layer. For Reacher, we use 3 convolutional layers followed by 3 fully connected layers. The 3D Teapot does not have an explicit encoder, i.e. it is the identity function. The output of the non-equivariant encoder is a 2-dimensional vector for 2D Shapes, 3D Blocks, and Rush Hour and a 4-dimensional vector for Reacher. The output of the equivariant encoders for each environment is listed in Table 1.

**Transition** $T$    The object-oriented environments use GNN transition models where the edge and node networks have the same structure as the encoder (3-layer MLPs). For Reacher and 3D Teapot, the transition model $T$ is a MLP with 512 hidden units for the non-equivariant version and 256 for the equivariant version. Actions are concatenated to the latent $z$ and are input into the transition models which then outputs a $z'$ of the same dimension as the input $z$. We use one-hot encoding for discrete actions.

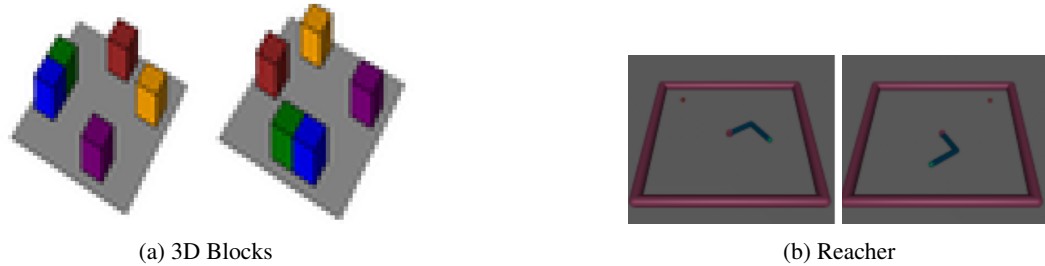

(a) 3D Blocks                                    (b) Reacher

Figure 7: Original observations and their $G$-transformed versions

**Gram-Schmidt embedding for Teapot transition model** In the case of the teapot domain, the transition model is constrained to output an element of $SO(3)$ representing a positively-oriented orthonormal frame. This is achieved by having the network output two vector $u, v \in \mathbb{R}^3$ and then performing Gram-Schmidt orthogonalization. Only two vectors are necessary since orthogonality and orientation determine the third, after producing two orthonormal vectors $u', v'$, the third vector $w'$ is uniquely determined by the property that it completes a positively-oriented orthonormal frame and can be computed by cross product. In summary,

$$u' = u/\|u\|, \qquad v' = \frac{v - (u' \cdot v)u'}{\|v - (u' \cdot v)u'\|},$$
$$w' = u' \times v', \qquad y = [u'\ v'\ w'].$$

### D.3 Datasets and Hyperparameters

For training, we use 1000 episodes of length 100 as training data for the grid world environments (2D shapes, 3D blocks, Rush Hour), 2000 episodes of length 10 for Reacher, and 100,000 episodes of length 1 for the 3D teapot. For Reacher, the starting state is restricted to a subset of the whole state space, so we perform warm starts with 50 random actions in order to generate more diverse data. The evaluation datasets are generated with different seeds from the training data to ensure that transitions are different.

For the object-oriented environments, we follow the hyperparameters used in (Kipf et al., 2020): a learning rate of $5 \times 10^{-4}$, batch size of 1024, 100 epochs, and the hinge margin $\gamma = 1$. We find that these hyperparameters work well for all other environments, except that Reacher uses a batch size of 256 and mixed precision training was used for both non-equivariant, fully-equivariant, and our method, in order to keep the batch size relatively high for stable contrastive learning. Most experiments were run on a single Nvidia RTX 2080Ti except for 3D Cubes which used a single Nvidia P100 12GB.

## E Group Representations

We explain the notation and definitions of the different representations of the groups considered in the paper and displayed in Table 1.

The $\rho_{\text{std}}$ representation of $C_4$ or $D_4$ on $\mathbb{R}^2$ is by 2-by-2 rotation and reflection matrices. The $\rho_{\text{std}}$ representation of $S_5$ permutes the standard basis of $\mathbb{R}^5$. The regular representation $\rho_{\text{reg}}$ of $G$ permutes the basis element of $\mathbb{R}^{|G|}$ according to the multiplication table of $G$. The trivial representation of $G$ fixes $\mathbb{R}$ as $\rho_{\text{triv}}(g) \cdot x = x$. For $D_4$, $\rho_{\text{flip}}(g) = \pm 1$ is a representation on $\mathbb{R}$ depending only on if $g$ contains a reflection. Given representations $(\rho_1, \mathbb{R}^{n_1})$ and $(\rho_2, \mathbb{R}^{n_2})$ of $G_1$ and $G_2$, $(\rho_1 \boxtimes \rho_2)(g_1, g_2)(v \otimes w) = g_1 v \otimes g_2 w$ gives a representation on $G_1 \times G_2$ on $\mathbb{R}^{n_1} \otimes \mathbb{R}^{n_2}$.

