# OpenReview forum: "Learning Symmetric Representations for Equivariant World Models"
_ICLR.cc/2022/Conference — ICLR 2022 Submitted_

### Official Review · Reviewer_LG1G · 2021-11-03

**Correctness:** 2
**Technical Novelty And Significance:** 2
**Empirical Novelty And Significance:** 2
**Recommendation:** 6
**Confidence:** 3

**Main Review:**

I found this paper incredibly dense in parts, and the presentation quite confusing. The paper mixes a variety of interesting but sometimes disparate topics - equivariance as it applies to networks, the concept of symmetry as it applies to group actions, and equivariant transitions. In a sense the authors might be trying to sell too much here, and as a result the actual contributions are quite hard to tease out. The quality of writing is also a concern. To the extent that I follow the arguments, the main strength is the richness of the problems considered here - equivariance is indeed a hot topic, and the general goal of trying to recover representations that capture aspects of symmetries of the underlying phenomena (e.g. in the latent space) when from observations such as visual images, these symmetries are not easy to tease out (e.g. the rotating car example in Figure 1.). The experimental results do appear to be promising, although the use of some of the metrics (e.g. DIE) are only relevant for the restricted case of the Euclidean groups or its sub-groups.

Weaknesses of the article include the quality of presentation which could be vastly improved, e.g, the related work section is rather disorganized - there doesn't seem to be a common thread. Simplifying the claims and accordingly the presentation, would add focus to the paper. There is also a rather a strong limitation of their assumption that for the task at hand the relevant groups are known. For particular cases there could be much more straightforward solutions. For example, Quessard et al. in the article "Learning Group Structure and Disentangled Representations of Dynamical Environments" (NeurIPS 2020) offer a much cleaner solution where a group G acts on pixel space, in such a way that the operations can be captured by matrix multiplication. The basic formulation in the Quessard et al. paper could certainly be applied to the teapot task it seems, one doesn't need the complexity of the equivariant transitions in Section 4.2 of the present paper to do this.


**Summary Of The Paper:**

This article assumes knowledge of an underlying group to learn symmetric latent representations, by utilizing equivariant transition models. The paper claims that the basic setup can be applied to a variety of existing equivariant networks (or non-equivariant ones) to improve performance as well as data efficiency. Experimental results are shown on 3 tasks, which are somewhat hand-crafted, in that for each of these tasks the allowed groups are assumed to be known a priori. Some of these experimental results are intriguing but the results themselves beg the question of why one needs all the complex machinery, at least in comparison with other recent efforts which admit a more direct mechanism (e.g. with regard to SO(3) and the teapot rotation task - see comments below).

**Summary Of The Review:**

A lot is packed in to the paper, making the message a bit confusing, and the main contribution not so easy to tease out. Further, it isn't clear to me what the theoretical guarantees are and it seems that a fair amount has to be known to apply the methods and to effect the equivariant transitions. To me it seems that simpler solutions do exist, at least for some cases, but perhaps I am missing something here. I also found the symmetric embedding steps in the Meta-Architecture to be somewhat engineered. The authors are indeed grappling with some interesting and very relevant questions, but separating the theory from what is engineered is not so easy (for this reviewer at least).

---

> ### Author Response · Authors · 2021-11-19
> **Response to specific questions**
>
> >I found this paper incredibly dense in parts, and the presentation quite confusing. The paper mixes a variety of interesting but sometimes disparate topics - equivariance as it applies to networks, the concept of symmetry as it applies to group actions, and equivariant transitions. In a sense the authors might be trying to sell too much here, and as a result the actual contributions are quite hard to tease out. The quality of writing is also a concern.
>
> We see our paper as a combination of two elements: (a) learning with partially equivariant models and (b) learning the transition dynamics of a Markov Decision Process. Both equivariance as it applies to networks and the concept of symmetry as it applies to group actions fall under (a). We use the concept of symmetry and group actions to derive the particular form of the equivariant networks.  For example, the 3D blocks world has a 90-degree rotational symmetry, which can be described in terms of the action of the discrete rotation group $C_4 = \lbrace 1, rot(90), rot(180), rot(270) \rbrace$ by saying that the dynamics of the world are invariant to $C_4$ rotational transformations.  Then the representation theory of $C_4$ can be used to construct equivariant layers for an equivariant neural network.
>
> > The experimental results do appear to be promising, although the use of some of the metrics (e.g. DIE) are only relevant for the restricted case of the Euclidean groups or its sub-groups.
>
> DIE does assume that the transformation is distance preserving and the other metric EE assumes the input is spatial. However, it is very difficult to accurately measure the level of equivariance of a non-equivariant latent space since there is no reasonable group action. In order to evaluate all models fairly, we make some assumptions in the evaluation metrics.
>
> > Weaknesses of the article include the quality of presentation which could be vastly improved, e.g, the related work section is rather disorganized - there doesn't seem to be a common thread. Simplifying the claims and accordingly the presentation, would add focus to the paper.
>
> We have significantly reorganized the paper including the related work and methods section, and polished the presentation. We hope that this revised draft makes our approach and contributions clear.
>
>
> >There is also a rather a strong limitation of their assumption that for the task at hand the relevant groups are known.
>
> This is a common assumption for equivariant networks, which operate over many useful domains. Our method has an even weaker assumption in that we assume the “simpler” group action $\rho_Y$ is known, while the symmetry group at the pixel-level $\rho_S$ is unknown. This enables even wider applicability. We combine the power of ENNs with standard networks and demonstrate that ENNs can act as an inductive bias to the non-equivariant network.
>
> > For particular cases there could be much more straightforward solutions. For example, Quessard et al. in the article "Learning Group Structure and Disentangled Representations of Dynamical Environments" (NeurIPS 2020) offer a much cleaner solution where a group G acts on pixel space, in such a way that the operations can be captured by matrix multiplication. The basic formulation in the Quessard et al. paper could certainly be applied to the teapot task it seems, one doesn't need the complexity of the equivariant transitions in Section 4.2 of the present paper to do this.
>
> It is true that the method in Quessard 2020 can be applied to some cases, such as the teapot task. However our method is more general in that the actions of the MDP are not the same as the symmetry group, and this is true for many commonly used environments. Furthermore, their method restricts the latent space to be linear with transition given by a matrix multiplication, while ours does not.  Our latent space transition model is given by a trainable equivariant neural network.  Consider the 3D cubes environment.  Our model learns an interpretable latent state consisting of the (x,y) coordinates of all the blocks and their transitions which include collisions. Modeling these complex dynamics with matrix multiplication would require a much higher dimensional latent space at best and would probably be impractical.

---

> > ### Comment · Reviewer_LG1G · 2021-11-29
> > **Response to reviewers**
> >
> > I apologies for the delay. It took me a while to get through the revised paper and also to follow the discussion on this paper. I'd like to thank the authors for considering comments received, and for the detailed response to my comments. I do think this is an interesting approach but there are some limitations, as acknowledged in the response. I do believe  that the requirement that the group for a task be known is a limitation; I'm not sure I agree with the argument in defense of this. I also don't fully see the generality of the approach -- see also the concerns raised by reviewer Uu3z. I do appreciate that the presentation has improved quite a bit since the first iteration. The main ideas might now be a bit more accessible to the wider community. I'm willing to raise my rating a bit, but I'm still on the fence on this article. Minor comment - in the background/relevant work section you mention "the teapot" task, which is a bit out of context since the task is only defined later in the article. Perhaps it would be better to just mention the general type of task at this stage.

---

### Official Review · Reviewer_Uu3z · 2021-11-03

**Correctness:** 3
**Technical Novelty And Significance:** 2
**Empirical Novelty And Significance:** 2
**Recommendation:** 6
**Confidence:** 4

**Main Review:**


The goal of the paper in using equivariance in latent space models in RL is interesting and motivated. Another positive point about the paper is that it evaluates its proposal on diverse problems. However, I have questions and concerns about the claims of contribution, the basic setup, theoretical presentation, and experimental results of this paper.

Contributions
------------------
The paper claims that “the idea of learning symmetric embedding networks has, to our knowledge, not previously been proposed or demonstrated.” and this is discussed as one of the main contributions. There have been several works in this area in recent years, including using symmetries in latent space for dynamical environments; e.g.,

- Higgins, Irina, et al. "Towards a definition of disentangled representations." arXiv preprint arXiv:1812.02230 (2018).
- Quessard, Robin, Thomas D. Barrett, and William R. Clements. "Learning group structure and disentangled representations of dynamical environments." arXiv preprint arXiv:2002.06991 (2020).


The proposed model
---------------------------
In the general case, the symmetry transformation of the action should be state-dependent. This is noted in prior work such as the one cited below. The theoretical setup used in the paper is therefore inadequate for symmetric environments. Generally, there is no discussion on prior works on symmetric MDPs and the way this work fits in.

- van der Pol, E., Worrall, D., van Hoof, H., Oliehoek, F., & Welling, M. (2020). MDP homomorphic networks: Group symmetries in reinforcement learning. Advances in Neural Information Processing Systems, 33.

Also, the motivation for having an equivariant encoder "after" a non-equivariant one is not clear, since the embedding will not be equivariant. This is especially so given that the equivariance of the representation is not enforced in any other way. My understanding is that the loss function used here only ensures that the transition in the latent space is consistent with the state space - i.e., the loss is exactly the same as that of Kipf et al (2019), am I correct?

Experiments
---------------
While the diversity in the choice of environments is great and the evaluation metric is reasonable, it is difficult to read much from the results. There are also some baselines that are difficult to justify: why use an equivariant network with a particular assumption of linear action on the input space where we know the group action is non-linear? This is further complicated by the fact that such a baseline is producing good results. With the exception of visualizations in the appendix (which are supportive), the results are mixed, borderline, or difficult to interpret.



**Summary Of The Paper:**

The paper proposes to learn latent representation and transition models that are equivariant to symmetry transformations of states and actions. The proposed “meta-architecture” consists of an encoder followed by an equivariant encoder and an equivariant transition model. The model is trained using (state, action, state) triples, where contrastive loss is used to prevent the collapse. The paper evaluates the proposed model in four different settings. In each case, a different choice of symmetry group and equivariant architecture is used. The quality of the learned latent representation is evaluated by comparing the relative distance of the predicted next state embedding to the true next state embedding, in comparison to negative samples.


**Summary Of The Review:**

see above

---

> ### Author Response · Authors · 2021-11-19
> **Response to specific questions and concerns**
>
> > The paper claims that “the idea of learning symmetric embedding networks has, to our knowledge, not previously been proposed or demonstrated.” and this is discussed as one of the main contributions.
>
> Thank you for pointing this out. Our wording was overly broad, and we have revised the relevant sentences.  Though the cited works learn symmetric embeddings, they do not employ equivariant neural networks.  The claim is now “The idea of learning symmetric embeddings using a mix of standard and equivariant networks has, to our knowledge, not previously been proposed or demonstrated.” Even though the symmetric embedding network is not equivariant by construction, the downstream equivariant networks allow it to learn a symmetric representation although $\rho_S$ is not explicitly known.
>
> > In the general case, the symmetry transformation of the action should be state-dependent. This is noted in prior work such as the one cited below. The theoretical setup used in the paper is therefore inadequate for symmetric environments.
>
>  While this is true in the most general case, many commonly used environments, the domains used in our experiments, and the ones used in van der Pol et al. (2020) do not require state-dependent transformations on the actions.  van der Pol et al. do consider this case in their formulation but not in their experiments (see Table 1 of their paper). Note that our method can easily accommodate state-dependent action transformations by enforcing equivariance in the transition model to the joint group action on the state-action product space $\mathcal{S} \times \mathcal{A}$. We clarify this assumption in the main text and include more details on the state-dependent formulation in the Appendix.
>
> > Generally, there is no discussion on prior works on symmetric MDPs and the way this work fits in.
>
> We have added discussion of several prior works on symmetric MDPs and their relation to our current work in the related work section.  van der Pol et al. (2020) uses the notion of MDP homomorphisms (Ravindran 2004) to build equivariant policy networks to reduce the size of the solution space. While we consider the same setting of MDPs with symmetry, we focus more on learning transition dynamics by building partially equivariant world models (and ignore the reward function) rather than learning equivariant policies.  We also provide a more detailed overview of MDP homomorphisms, symmetric MDPs, and their connection to learning equivariant world models in the Appendix.
>
> > the motivation for having an equivariant encoder "after" a non-equivariant one is not clear, since the embedding will not be equivariant.
>
> The equivariance of the representation is not enforced in any other way and the loss function is the same as that of Kipf et al (2019). We make the point that no special structure is needed to learn equivariant representations besides that the downstream networks which are equivariant by construction can encourage non-equivariant networks to learn such representations.  The purpose of the symmetric embedding network $S$ is to learn an equivariant representation whereas the purpose of the encoder $E$ is to perform dimensional reduction in an equivariant manner.
>
> > There are also some baselines that are difficult to justify: why use an equivariant network with a particular assumption of linear action on the input space where we know the group action is non-linear?
>
> There is a spectrum of possible models that utilize equivariance and we use the two ends of the spectrum (non-equivariant and fully equivariant models) as baselines to evaluate our partially equivariant approach.  Surprisingly, despite the fact that the inductive bias for the equivariant network is overly strong, the fully equivariant model performs well even in this setting, a result that requires further investigation. We have some hypotheses on why.    Consider the Reacher environment.  Due to the skewed perspective, we can see that the simple pixel-level rotation maps training data to out-of-distribution images with a different skewed perspective which are never seen by the model.  We hypothesize that the equivariance constraint thus does not hamper the model’s performance on training data, but only constrains its extrapolation capabilities to out-of-distribution samples, which neither helps nor hurts the model during evaluation.
>
> However, in the case we train with limited actions, the fully equivariant model must use equivariance to group transformations to extrapolate to new actions.  In this case (see Table 5 and Table 6) the hard-coded but incorrect pixel-level transformation results in worse performance than our method as we would expect.  We have added further explanation to the paper.

---

> > ### Comment · Reviewer_Uu3z · 2021-11-22
> > **Follow up**
> >
> > Thank you for your comments and also for updating the paper based on reviewer feedback.
> > Let me elaborate on my concerns with the last two items. I believe your response didn't help as much with these items:
> >
> > - regarding architecture and the loss: consider your response about the loss, I can't see why adding equivariant layers after non-equivariant ones "can encourage non-equivariant networks to learn [equivariant] representations". For example, if you put a convolution layer after a fully connected layer does the convolution layer encourage the fully connected layer to become translation-equivariant? If this is indeed the case it would be very interesting to show.  This would serve as an independent contribution that can have interesting applications outside RL.
> >
> > - regarding experiments: I understand your objective in considering two ends of the spectrum in the study of equivariance. However, the fully equivariant end is equivariant to a wrong set of transformations. This undermines your original objective in the design of the experiment. Ideally, you should choose an example in which the symmetries of the data match the equivariance constraint of the model. Additionally, using the model that is biased using the wrong symmetry assumption you are reporting a better performance than the model that does not have that bias.

---

> > > ### Author Response · Authors · 2021-11-24
> > > **Response to follow up [1/2]**
> > >
> > >  > regarding architecture and the loss: consider your response about the loss, I can't see why adding equivariant layers after non-equivariant ones "can encourage non-equivariant networks to learn [equivariant] representations".
> > >
> > > In [1], the authors use a non-equivariant network (“object extractor” akin to our symmetric embedding network $S$), a non-equivariant encoder, and a permutation equivariant (with respect to objects) graph neural network for the transition function and train the model end-to-end using a contrastive loss (we use the same loss). Their results on 2D shapes and 3D block environments (Table 1) show that the non-equivariant extractor learns to isolate each object and thus learns permutation equivariance. We obtain a similar result for object permutations and additionally consider rotations for these environments.
> > >
> > > Our intuition behind why this works is that the ground truth function is equivariant and so any highly accurate model must learn this equivariance. By enforcing equivariance on part of the model, we thus assist the non-equivariant model components to learn equivariance.
> > >
> > > The experiment you propose is a great suggestion and would definitely have broader consequences beyond RL. In a toy MNIST classification experiment, we construct a 3-layer network (a linear followed by convolutional and then a linear layer) and train using cross entropy. We find that for small translations in the input image (up to $10\\%$ of image size), the classification probabilities change less than $\pm 5\\%$ on average. This suggests that the first linear layer learns approximately local translation equivariance.
> > >
> > > [1] Kipf, Thomas, Elise van der Pol, and Max Welling. "Contrastive learning of structured world models." ICLR 2020.
> > >
> > > > regarding experiments: I understand your objective in considering two ends of the spectrum in the study of equivariance. However, the fully equivariant end is equivariant to a wrong set of transformations. This undermines your original objective in the design of the experiment.  Ideally, you should choose an example in which the symmetries of the data match the equivariance constraint of the model.
> > >
> > > We consider the setting where $\rho_S$ is unknown or hard to compute, which is broadly true for many environments. Within this setting, the pixel-level symmetries of the fully equivariant model are indeed wrong, but as $\rho_S$ is unknown, they are the closest pixel-level symmetries to the true transformation property and should be considered as approximations rather than simply incorrect symmetries. One could ask if the equivariance error of using pixel-level rotations is a small enough price to pay for having strictly imposed end-to-end equivariance (which we cannot have if we are using the true $\rho_S$).
> > >
> > > Furthermore, as we point out in our response and in the revised paper (Sec. 5.4), imposing equivariance to approximately correct transformations does not necessarily force the model to be inaccurate, since the images transformed by the incorrect pixel-level symmetries are never seen in the training data. In fact, if the pixel-level symmetries are close to the true ground truth input symmetry, this may even help the model somewhat to learn the correct output.
> > >
> > > We give a simple example to explain this point. Given $f: X \to Y$, let $\rho_{S}$ and $\rho_{\mathrm{pixel}}$ be the true and approximate pixel-level symmetry transformations, respectively.  We impose $f$ to be equivariant to $\rho_{\mathrm{pixel}}$. A sample $x$ is transformed to $x’$ w.r.t. $\rho_{S}$, which is within the training distribution, and to $x^{\mathrm{pix}}$ w.r.t $\rho_{\mathrm{pixel}}$, which never appears in the training data.  The model is then only constrained to generalize to $x^{\mathrm{pix}}$, but this is irrelevant since $x^{\mathrm{pix}}$ is out of distribution. Meanwhile, the imposed constraint does not limit the model in learning $x’$ correctly.
> > >
> > > If we assume that $\rho_S$ is known, then logically utilizing this knowledge would be best.   This follows the usual finding for equivariant neural networks (see e.g. [2]) that imposing a strict symmetry bias with a known group is preferable to learning the underlying group. Note that a fully equivariant model where $\rho_S$ is given would then act as an oracle, using information that is not given to the other models. For 2D Shapes and Rush Hour, the pixel-level symmetries are in fact the correct transformation (and thus these environments are toy cases for us). In these cases, the fully equivariant model achieves very low values on the equivariance metrics, as expected.
> > >
> > > [2] Dehmamy, N. , Walters, R., Liu,Y., Wang, D., Yu, R. “Automatic Symmetry Discovery with Lie Algebra Convolutional Network”. NeurIPS 2021.

---

> > > > ### Author Response · Authors · 2021-11-24
> > > > **Response to follow up [2/2]**
> > > >
> > > > > Additionally, using the model that is biased using the wrong symmetry assumption you are reporting a better performance than the model that does not have that bias.
> > > >
> > > > Tables 2 and 3 show that the fully equivariant model has similar performance to ours. In the limited actions generalization experiments (Tables 5 and 6), it is impossible for the fully equivariant model to learn the transformed input $(\rho_{S}(g)x)$ separately and thus our method has better performance. This aligns with our hypothesis above and demonstrates the advantage of our method.

---

> > > > > ### Comment · Reviewer_Uu3z · 2021-11-29
> > > > > **Response**
> > > > >
> > > > > Thank you for your comments. I appreciate the changes made to the paper and the efforts in responding to reviews. However, my questions/ concerns remain, let me elaborate :
> > > > >
> > > > > 1- your architecture assumes that adding equivariant layers after non-equivariant ones "can encourage non-equivariant networks to learn [equivariant] representations". The paper provides no evidence for this claim. Your most recent comments state that prior work tries to achieve equivariance to permutation simply using MLP and relying on the data. Therefore your choice of using an equivariant layer after an equivariant one can only improve things. However, the first statement does not support the latter. The former is a correct statement, with increasing amount of training data an MLP will eventually become equivariant. However, I can't find any evidence for the latter, in the paper or in your comments.
> > > > >
> > > > > Regarding MNIST experiment: why not directly test the hypothesis that putting an equivariant layer after a fully connected one helps with equivariance? The result reported here doesn't lead to the conclusion you want -- it is only showing that doing so doesn't do much damage for classification. Did I miss anything?
> > > > >
> > > > > 2. RE: "consider the setting where is unknown or hard to compute, which is broadly true for many environments. Within this setting, the pixel-level symmetries of the fully equivariant model are indeed wrong, but as is unknown, they are the closest pixel-level symmetries to the true transformation property and should be considered as approximations rather than simply incorrect symmetries"
> > > > >
> > > > > At the risk of repeating myself, this is not the right way of testing your network against an equivariant network. As suggested in my earlier comment a correct approach here would be to pick a fully equivariant example. I did not quite follow the line of reasoning in your response but I believe your recent response argues that imposing equivariance to wrong symmetries does no harm (maybe I misunderstood?) I disagree with this statement. Imposing symmetry constraints that are non-existent in the data on the mode deteriorates its performance. Consider using a ConvNet for a feature vector that has no translation symmetry. This can significantly undermine the performance of the model. The experiments of the paper that compare against an equivariant network (with wrong symmetry specification) are of such nature.

---

> > > > > > ### Author Response · Authors · 2021-11-30
> > > > > > **Clarification**
> > > > > >
> > > > > >
> > > > > > Thank you for following up and continuing the conversion.  We really appreciate your time and questions.
> > > > > >
> > > > > > >Your most recent comments state that prior work tries to achieve equivariance to permutation simply using MLP and relying on the data.
> > > > > >
> > > > > > Sorry if we weren’t clear. The prior work (Kipf et al., 2020) consists of non-equivariant layers followed by equivariant layers: a non-equivariant CNN+MLP is followed by a permutation-equivariant network (GNN).  Here, it is shown that the non-equivariant part of the model can separate individual objects regardless of their configuration; thus, learning to be equivariant to permutations.  The "-latent GNN" ablation which replaces the permutation-equivariant network with a non-permutation equivariant one does not perform as well or learn permutation-equivariance.
> > > > > >
> > > > > > > it is only showing that doing so doesn't do much damage for classification. Did I miss anything?
> > > > > >
> > > > > > We are not directly commenting on the classification accuracy, only how the outputs of the network change.  That is, we are directly measuring the invariance of the network $f: \mathbb{R}^{28 \times 28} \to \mathbb{R}^{10}$ and showing $|f(gx) - f(x)|$ is small for $g$ a translation.
> > > > > >
> > > > > > >As suggested in my earlier comment a correct approach here would be to pick a fully equivariant example.
> > > > > >
> > > > > > Sorry if we have misunderstood.  The Rush Hour and 2D Shapes environments are equivariant examples.  Here we compare to an equivariant network with the correct equivariance bias and see our method does similarly or only slightly worse.  We consider this a positive result since in these cases where the correct bias is known and can be used, it is better to do so than use our method to learn it.
> > > > > >
> > > > > > > I did not quite follow the line of reasoning in your response but I believe your recent response argues that imposing equivariance to wrong symmetries does no harm (maybe I misunderstood?) I disagree with this statement. Imposing symmetry constraints that are non-existent in the data on the mode deteriorates its performance. Consider using a ConvNet for a feature vector that has no translation symmetry. This can significantly undermine the performance of the model. The experiments of the paper that compare against an equivariant network (with wrong symmetry specification) are of such nature.
> > > > > >
> > > > > > We definitely agree that an incorrect equivariance bias can hurt model performance as in your example.  However, it does not *need* to hurt.  This is a subtle point.  In general, incorrectly constraining $f(gx) = gf(x)$ hurts if both $gx$ and $x$ are *in distribution*.  In our setting $gx$ is not in distribution.

---

> > > > > > > ### Comment · Reviewer_Uu3z · 2021-11-30
> > > > > > > **further clarification**
> > > > > > >
> > > > > > > Regarding Kipf et al (2020), I believe the equivariant GNN was the transition model not part of the embedding. I'm not sure to what extent this makes a difference. However, I see now that the error you are reporting is on the invariance. This is a positive sign, and I encourage you to include a similar experiment in the paper this time measuring the "equivariance" of the network. In light of this and your clarification on the second question as well, I'm happy to increase my score. However, I encourage you to include the main points of this discussion in the paper.

---

### Official Review · Reviewer_5ro3 · 2021-11-04

**Correctness:** 3
**Technical Novelty And Significance:** 1
**Empirical Novelty And Significance:** 3
**Recommendation:** 6
**Confidence:** 3

**Main Review:**

The presentation can be improved in several ways. I agree that the paper studies various models, symmetry assumptions, and different controlled environments so it is a challenge to make it self-contained. Still, there are several things that I found hard to understand:

- 'We consider the case of MDPs with symmetry as in van der Pol et al. (2020)'. Maybe I missed something but this model is not motivated anywhere.  is it obvious for computational purposes? it is not obvious to me why study/build on this model? The background section gives some motivation behind Kipf et al. and structured contrastive models but Van der Pol et. al 2020 is mentioned here for the first time.
How central is the MDP assumption in validation of the main idea? Is it reasonable to ask to demonstrate the main idea on other structured spaces?

-'We require that ρA(G) · A′ = A. That is, every MDP action is a G-transformed version of one in A′ .' this should be mentioned at the beginning of Model overview, Section 4.1.

The experiments are not completely convincing. In particular, fully equivariant networks that assume the correct transformation $\rho_S$  is a simple transformation of the pixels performs very well in all tests and thus, somewhat contradicts the main motivation of the paper.

The paper is riddled with typos, some mentioned below:
-Future work will include applying *out* method  - The MDP action *is* input directly to T. - 'each element $g \in G$ *must* invertible with respect to composition' .. We demonstrate that this meta-architecture can be *use* to learn world models with a variety. We then learn a transition model in latent space where *can* enforce symmetry

**Summary Of The Paper:**

This paper proposes to learn an equivariant world model without access to group representation on input space $\rho_S$. To this end, it learns a symmetric abstract state mapping from states $s$ to abstract states $z$ in a space $Z$ with an explicit action $\rho_Z$ of the symmetry group G. It then learns a transition model in latent space that can enforce symmetry using an equivariant neural network.

The problem of learning the equivariant model without assuming any group representation on input space is important and motivated well in the paper. The paper evaluates the model and compares it extensively on various benchmarks.

**Summary Of The Review:**

I give marginally above as the experimental setup seems rigorous. The authors are quite transparent about the lack of theoretical novelty when they state they are not proposing a new equivariant neural network design and instead, they build upon previous work to demonstrate working in new domains with unspecified group actions. I will revisit the score depending on other reviews. I can not comment on the utility of the idea only in MDP structured space but in general, the problem is relevent.

---

> ### Author Response · Authors · 2021-11-19
> **Response to specific questions and concerns**
>
> > 'We consider the case of MDPs with symmetry as in van der Pol et al. (2020)'. Maybe I missed something but this model is not motivated anywhere.
>
> We have added more details about the concept of MDPs with symmetries in the related work and Appendix to help clarify the connection.  Specifically, the task in our paper is to model an equivariant transition function, which is part of an MDP with symmetries.  (We omit the reward function.)  In this sense, our work fits within the conceptual framework of van der Pot et al. (2020) and before this, Ravindran (2004).   However, in their work, they focus on learning an optimal policy and assume knowledge of the group action $\rho_S$.
> In terms of validation, the fact we consider an MDP with symmetries allows us to perform experiments with held-out actions in the training set and still generalize to the full action set.
>
> > 'We require that ρA(G) · A′ = A. That is, every MDP action is a G-transformed version of one in A′ .' this should be mentioned at the beginning of Model overview, Section 4.1.
>
> This assumption is only required for the generalization experiments where we train only on a subset of the actions. We have clarified this point in the text.
>
> > The experiments are not completely convincing. In particular, fully equivariant networks that assume the correct transformation ρS is a simple transformation of the pixels performs very well in all tests and thus, somewhat contradicts the main motivation of the paper.
>
> Despite the fact that it enforces an incorrect inductive bias, the fully equivariant model does perform very well (except for Table 5, in which it performs significantly worse than ours.) We hypothesize that due to the skewed perspective, the simple pixel-level rotations would produce samples that are not seen in the training data, i.e. are out-of-distribution. The equivariant bias thus may not hinder learning the correct representations for in-distribution training data allowing it to achieve good performance, and merely enforces the model to be equivariant to unseen out-of-distribution samples. This is not true for the generalization from limited actions experiments as the fully equivariant model needs to rely on the correct group transformation in order to extrapolate to unseen actions. We see in Tables 5 and 6 that the fully equivariant model performs worse than our model as expected. We have added this hypothesis to the discussion in the paper.
>
> > The paper is riddled with typos
>
> Thank you for pointing these out. We have fixed these typos and reviewed the paper several times to bring it up to a higher level of polish.

---

### Official Review · Reviewer_hasv · 2021-11-08

**Correctness:** 3
**Technical Novelty And Significance:** 3
**Empirical Novelty And Significance:** 3
**Recommendation:** 6
**Confidence:** 2

**Main Review:**

I found this to be a very difficult paper to read and understand.  In part it is a new area for me, but also I feel the authors are writing to others who live in their own world rather than a broader community of researchers.

Figure 2 is helpful, but I could not understand what is the purpose of the symmetric embedding network S as opposed to the encoder E?  Why two networks instead of one?

Also, as far as I understand S, E and T are just standard neural networks, there is nothing about SO(3) per se or the mathematics of transformations engineered into them.  It appears the group theory mainly comes into play in how the networks are trained, as opposed to the computations they perform per se.  If so, then to me this seems a shame because I think where group theory could help us the most is in moving away from standard MLP models to more powerful and interesting computational frameworks.

The tables 2-6 showing model performance are also very hard to understand - extremely cryptic.   A bit more information about what these abbreviations mean and what is being measured would be helpful.


**Summary Of The Paper:**

The goal of this paper is to learn world models - where the network must predict actions from images - in an efficient manner.  This is done by exploiting symmetries in the data.   Results are shown for a number of world model benchmarks, showing improvement over standard methods.


**Summary Of The Review:**

An interesting idea but a difficult paper to read and understand.

---

> ### Author Response · Authors · 2021-11-19
> **Response to specific questions and concerns**
>
> > I found this to be a very difficult paper to read and understand.
>
> We have made significant revisions to the paper to improve clarity and presentation. See the general response for a summary.
>
> > Figure 2 is helpful, but I could not understand what is the purpose of the symmetric embedding network S as opposed to the encoder E?
>
> Good question.  The composite function $z = E \circ S(x)$ is intended to perform two functions: creating a symmetric embedding and reducing the dimension.  Our intention is these roles be split across $S$ and $E$. The symmetric embedding network $S$ transforms the pixels into a latent intermediate code. It is not explicitly equivariant. The encoder $E$ is an explicit equivariant network that maps the intermediate code into a lower dimensional vector in order to distill only the relevant features. The intuition behind these different networks is that the symmetric embedding network $S$ learns a nonlinear transformation from the input space to an intermediate space with a learned representation $\rho_Y$, from which the equivariant encoder can distill relevant features into a lower-dimensional latent space.
>
> > Also, as far as I understand S, E and T are just standard neural networks, there is nothing about SO(3) per se or the mathematics of transformations engineered into them. It appears the group theory mainly comes into play in how the networks are trained, as opposed to the computations they perform per se.
>
> This is a misunderstanding.  In our method, the symmetric embedding network S is a standard CNN and so the symmetry is not engineered into it, but the encoder E and transition T are equivariant neural networks which are explicitly constrained by weight sharing to enforce the relevant symmetry.  We demonstrate that even though S is not equivariant by construction, it learns to produce symmetric intermediate representations with respect to symmetry group action explicitly imposed on the downstream models $E$ and $T$.  We have edited the text to make this clearer.
>
> > The tables 2-6 showing model performance are also very hard to understand - extremely cryptic. A bit more information about what these abbreviations mean and what is being measured would be helpful.
>
> We have revised the section on experiments for clarity. Several evaluation metrics were used in order to more comprehensively measure the level of symmetry of the learned latent representations, even for the non-equivariant baseline which uses standard neural networks for $S$, $E$, $T$.
> Hits and mean reciprocal rank MRR are used because there is no reconstruction for comparison to the ground truth, so it is hard to measure whether the latent representation is correct with respect to the image input. Thus we resort to ranking metrics as in [1].  $EE(f)$ is the equivariance error of the function f.  However, to measure $EE=|f(gx) - g(f(x))|$ requires a predefined group action on the input and output space of f which may be ill-defined for a non-equivariant model f. We develop distance invariance error DIE using the invariance of the loss to elements of the symmetry group. We apply it to the non-equivariant portion of our system, S, and to the whole model as well.
>
> [1] Thomas Kipf, Elise van der Pol, Max Welling. “Contrastive Learning of Structured World Models.” http://arxiv.org/abs/1911.12247

---

### Author Response · Authors · 2021-11-19
**Summary of Revisions0**

We thank the reviewers for their detailed comments. The reviewers agree that learning symmetries in latent space without knowledge of the group action on the input space is an important and valuable research direction. We have substantially revised the paper to address concerns about the overall structure and presentation of the paper, specific design choices of our method, relation to previous work, and the takeaways from the experimental results. For reviewers' convenience, we have highlighted all changes from the original submission in blue.

Summary of revisions:
1. Significantly reorganized and rewritten sections for better presentation.
2. Revised the experiment setup, results tables, and the discussion to make them easier to interpret and added a hypothesis for the fully equivariant model.
3. Clarified the role and structure of the Symmetric Embedding Network $S$, Encoder $E$ and Transition Model $T$.
4. Added references to related work by Higgins et al. 2019 and Quessard et al. 2020 and clarified their connections to the paper.
5. Added additional background on symmetric MDPs (Ravindran 2004, van der Pol et al. 2020) and equivariant world models and their relation to our method in Appendix B.
6. Made the assumption of $\rho_A(G) \cdot A^{′} = A$ more clear.

---

### Decision · Program_Chairs · 2022-01-20

**Decision:**

Reject

**Comment:**

This paper proposes a new method for learning symmetric representations for equivariant world models. All reviewers recognized the interesting results in the paper. The reviewers have raised some concerns, which were not addressed well yet after the rebuttal. For example, Reviewers LG1G and Uu3z mentioned about the limitation of using the group for a task and the generality of the approach. Reviewer 5ro3 mentioned about the lack of novelty. Though they gave 6, they were quite neutral about the paper acceptance. Eventually, after a second round of deiscussions, we had to make this difficult decision: The current form of this paper is not ready for publications.